# Determinants/Motivations of Corporate Social Responsibility Disclosure in Developing Economies: A Survey of the Extant Literature

**Waris Ali** [1,*]  **, Jeffrey Wilson** [2] **and Muhammad Husnain** [1]

1    Department of Business Administration, University of Sahiwal, Sahiwal 57000, Pakistan;
     m.husnain@uosahiwal.edu.pk
2    School of Environment, Enterprise and Development, University of Waterloo,
     Waterloo, ON N2L 3G1, Canada; jeffrey.wilson@uwaterloo.ca
*    Correspondence: waris.ali@uosahiwal.edu.pk

**Abstract:** The main purpose of this study is to systematically analyse and synthesise the empirical literature on the drivers and motivations of CSR disclosure in developing countries. Previous studies on CSR disclosure have primarily investigated the accuracy of disclosure claims, impact on various actors, and the factors deriving CSR disclosure. While literature on CSR disclosure dates back to 1983, the number of studies have increased substantially in recent years, with 86% of studies being published in the last decade and a half. The results revealed that both internal and external factors influence the disclosure of CSR information. Internal factors influencing CSR disclosure include company characteristics such as size, industry, financial performance, corporate governance elements such as board size and board independence, and types of ownership. In addition, corporate polices and concerns also influence the disclosure of CSR-related information. External category factors influencing CSR disclosure include, regulatory pressures, government pressure, media concerns, social-cultural factors, and industry-level factors such as the level of industry competition, customers' concerns, and multiple listing of a firm. Furthermore, global value chains, international buyers, international NGOs, and international regulatory bodies pressure companies in developing countries to disclose social and environmental information. In terms of motivations, companies disclose CSR information to improve their corporate reputation, improve their financial performance, access investment opportunities, and manage key stakeholders. The dominant theoretical frameworks used to explain the determinants of CSR disclosure include legitimacy theory and stakeholder theory.

**Keywords:** corporate social responsibility; CSR; motivations; environmental responsibility; social and environmental disclosure; developing countries; literature survey





## 1. Introduction

Although research on social and environmental disclosure dates back to 1983 [1], academic interest in the subject has grown significantly in the last two decades [2–4]. Studies attribute increased corporate social and environmental disclosure to numerous factors, such as corporate visibility, corporate governance processes, stakeholder pressures, and political, social and cultural concerns [3]. Several authors have previously reviewed the literature on CSR disclosure posing very interesting research questions [2–6]. While the existing reviews offer important insights, this study addresses four notable gaps. Firstly, the existing reviews have primarily focused on a small set of determinants of CSR disclosure, such as corporate governance, the CEO, and company characteristics [2,6–14]. Secondly, none of the existing assessments consider the motivations of CSR disclosure. Thirdly, the existing reviews on the determinants of CSR disclosure lack up-to-date information on the factors and consequences of CSR disclosure. Finally, the existing reviews used the narrative analysis technique to report their study findings [3,4,7,8,14]. The current study

takes a quantitative approach to investigate the factors and motivations of CSR disclosure in developing countries for the study period of 1983 to 2021. Based on a review of 71 empirical studies published in Chartered Association of Business Schools (ABS) ranked journals, this research aims to answer the following research questions:

(i)　　What is the current state of CSR disclosure research in developing countries?
(ii)　　What are the widely explored dimensions of CSR disclosure in developing countries?
(iii)　What are the underpinning theories of CSR disclosure research in developing countries?
(iv)　What are the measurements of CSR disclosure and its dimensions in developing countries?
(v)　　What are the determinants of CSR disclosure in developing countries?
(vi)　What are the motivations of CSR disclosure in developing countries?
(vii)　What are the avenues for future research?

The findings of this research will enable policymakers and corporate leaders to advance a corporate social responsibility agenda in developing countries. Furthermore, the authors propose a future research agenda to advance our understanding of CSR disclosure, focusing on developing countries, which are often neglected in CSR literature.

This paper is organised as follows. The section that follows discusses the methodology of this study. The main section reports the findings of the study. The final section presents the conclusion and future research directions.

## 2. Methodology

We used Denyer and Trandfield's [15] multi-step strategy to search the published articles on the determinants and motivations of CSR disclosure in developing countries. These steps are the following: (1) define the research questions; (2) establish the scope and boundaries; (3) identify, screen, and select studies; and (4) analyze and synthesize research findings.

### 2.1. Defining the Research Questions

The literature on CSR disclosure has grown substantially in the last four decades [2–4]. A comprehensive systematic literature review study focused on developing countries is strongly needed in order to assess the development of CSR disclosure, and its determinants and motivations over the last four decades. In the present study, we answer the research questions described in the introduction section.

### 2.2. Establishing the Scope and Boundaries of the Review

We selected studies based on several criteria in order to create a comprehensive database of CSR disclosure literature. To begin with, we chose studies that took place between 1983 and 2021. The year 1983 was chosen as the starting point because Singh and Ahuja published the first study describing the nature of CSR disclosure in 1983. As conceptual boundaries for this research, we used two key terms: determinants and motivations of CSR disclosure. Following this, we created a list of 12 keywords based on a review of the seminal papers in the field and combined them to form the search string for the present study. In order to improve the quality of the systematic literature review, this study included empirical work published in Association of Business School (ABS) ranked 2018 journals, while excluding books, book chapters, conference proceedings, and work published in predatory journals [16]. We searched articles in a variety of databases— including EBSCOhost, Web of Science (ISI), Elsevier Science Direct, SAGE Journals, Wiley Online Library, and Google Scholar—in order to create a comprehensive database of articles for the systematic literature review.

### 2.3. Identification, Screening, and Selection of studies

This step identifies, screens, and selects relevant studies for the review's purpose. We searched the databases noted above using the following keywords: determinants of CSR disclosures, consequences of CSR disclosures, determinants of social disclosures, consequences of social disclosures, determinants of environmental disclosure, consequences

of environmental disclosures, company characteristics and CSR disclosure, corporate governance and CSR disclosure, ownership structure and CSR disclosure, motivations of CSR disclosure, drivers of CSR disclosure, factors of CSR disclosure. The search resulted in 823 papers for consideration.

After uploading the identified articles into excel, the "remove duplication" command reduced the number of identified items from 823 to 421. The papers were then evaluated against the quality screening criteria, and those from ABS ranked 2018 journals (e.g., those ranked 1, 2, 3, 4, and 4*) were chosen. The Charted Association of Business Schools publishes an international journal ranking [17]. During the review selection process, this reduced the number of studies to 182. We further screened the 182 studies manually to ensure that respective studies addressed determinants and motivations of CSR disclosure in developing countries. To do so, we examined the abstracts, as well as the introduction and conclusion sections to select the final sample of 71 studies for evaluation. The selected sample are listed in Table 1.

### 2.4. Analysis and Synthesis

In order to avoid over-reliance on one study and under-reliance on others, the data from 71 investigations had to be clearly combined [17]. Narrative analysis can be used to examine a large body of literature [17,18]. Using narrative synthesis, we determined the context, theoretical perspectives, determinants, and motivations of CSR disclosure, as well as its aspects. We created separate sheets to record the drivers and effects of CSR disclosure, and reviewed them for possible inaccuracies [19]. These sheets aided us in compiling tables of theoretical perspectives, determinants, and motivations.

Furthermore, we performed an in-depth analysis of the findings in order to categorise them into factors, enabling the results to provide valuable insights for future research. This task was challenging due to the complexity of the field in terms of the theoretical perspectives used and the nature of the determinants and motivations reported. Therefore, we used a suitable framework to link our research questions to communicate the results logically. This framework offers readers a comprehensive understanding of the determinants and motivations of CSR disclosure.

**Table 1.** Overview of the extant literature.

| Study | Nature of Study | | Country | Theory | Outcomes | |
|---|---|---|---|---|---|---|
| | Determinants | Motivations | | | Antecedents | Motivations |
| [20] Singh and Ahuja (1983) | ✓ | | India | N/A | Firm size (+), industry (+), financial performance (+) | |
| [21] Teoh and Thong (1984) | ✓ | | Malaysia | N/A | Firm size related to commitment to social reporting (+), foreign ownership related to commitment to social reporting (+) | |
| [22] Maheshwari (1992) | ✓ | ✓ | India | LT | Firm size (+), industry (+) profitability (+), governmental pressures (+), market forces (+), community involvement (−) | Enhanced corporate profitability and social responsibility (+), fair business practices (+) |
| [23] Williams (1999) | ✓ | | Asian-pacific nations | PE | Culture, political, social system (+) | |
| [24] De-Villiers (2003) | ✓ | | South Africa | N/A | Absence of legal requirements (−), non-availability of data (−), lack of motivation for CSR disclosure (−) | |
| [25] Haniffa and Cooke (2005) | ✓ | | Malaysia | LT | Firm size (+), industry size (+), multiple listing (+), financial performance (+), culture proxied by Malay directors (+), governance structure (+) | |
| [26] Yusoff et al. (2006) | ✓ | ✓ | Malaysia | AT, LT and ST | Firm size (+), industry (+), environmental performance (+), financial performance (+), environmental expenditures (+), financing for environmental equipment (+) | More visibility of corporate environmental performance (+), enhance motivation to develop environmental management system (+) |
| [27] Alsaeed (2006) | ✓ | | Saudi-Arabia | N/A | Firm size (+), industry size (0), financial performance (0), firm age (0), creditors i.e., leverage (0), audit firm size (0), ownership dispersion (0) | |
| [28] Amran and Devi (2007) | ✓ | | Malaysia | PE | Influence of government proxied by govt. shareholdings (+), dependence on government (+) | |

**Table 1.** *Cont.*

| Study | Nature of Study | | Country | Theory | Outcomes | |
| --- | --- | --- | --- | --- | --- | --- |
| | Determinants | Motivations | | | Antecedents | Motivations |
| [29] Kamla (2007) | ✓ | | Middle East | N/A | Country specific factors (+) resulted in variation in themes of disclosure | |
| [30] Mirfazil (2008) | ✓ | ✓ | Indonesia | LT | Firm size (0), industry (+), transparent information (+), environmental performance (+), regulatory pressure (+), environmental concerns (+), stakeholder's concerns (+) | Adoption of the processes that are fulfilling the market demand (+), greater influence of firm's operations on stakeholders as well as shareholders (+) |
| [31] Amran and Devi (2008) | ✓ | | Malaysia | IT | Firm size (+), industry size (+), influence of government proxied by govt. shareholdings (+), dependence on govt. (+) | |
| [32] Wanderley et al. (2008) | ✓ | | Emerging Countries | N/A | Country (+) | |
| [33] Rizk et al. (2008) | ✓ | | Egypt | N/A | Industry size (+), ownership structure (+) | |
| [34] Mitchell and and Hill (2009) | ✓ | | South Africa | N/A | Industry size (+), absence of legal requirements (−), lack of motivation for disclosure (−), non-availability of data (−), cost of obtaining data (−) | |
| [35] Sobhani et al. (2009) | ✓ | | Bangladesh | N/A | Firm size (+), industry size (+), financial performance (+) | |
| [36] Hassan and Harahap (2010) | ✓ | ✓ | Indonesia | ST | Firm size (o), board size (+), corporate governance (+), stakeholder's concerns (+), environmental concerns (+), increased strategic social investments (+) | Enhance environmental protection using recyclable and environment friendly supplies (+); fair dealing with supply chain (+) |
| [37] Buniamin (2010) | ✓ | | Malaysia | LT | Industry size (+) | |
| [38] Huang and Kung (2010) | ✓ | | Taiwan | ST | Firm size (+), financial performance (+), government. (+), creditors i.e., leverage (−), consumers (+), suppliers (−), competitors (+), employees (+), shareholding concentration (−) | |

**Table 1.** *Cont.*

| Study | Nature of Study | | Country | Theory | Outcomes | |
|---|---|---|---|---|---|---|
| | Determinants | Motivations | | | Antecedents | Motivations |
| [39] Khan (2010) | ✓ | | Bangladesh | LT | Non-executive directors on board (+), foreign nationalities on board (+), women representation on board (0) | |
| [40] Saleh et al. (2010) | ✓ | | Malaysia | N/A | Firm size (+), financial performance (0), institutional ownership (+) | |
| [41] McCuinness et al. (2017) | ✓ | | China | CMT | Female CEO (+), female chairman of board (+), independent directors on board (0), CEO duality (0), board size (+), managerial size (+), managerial ownership (0), state ownership (−) | |
| [42] Mahadeo et al. (2011) | ✓ | | Mauritius | LT | Firm size (+), leverage (+) related to HR and ED | |
| [43] Abd-Rahman et al. (2011) | ✓ | | Malaysia | N/A | Firm size (+) | |
| [44] Qadan and Suwaidan (2019) | ✓ | | Jordan | AT | Board gender diversity (0), board size (+), board independence (−), CEO duality (−), director Age (−), ownership concentration (−), institutional ownership (−), foreign ownership (0) | Corporate accountability (+) |
| [45] Haji (2013) | ✓ | | Malaysia | LT | Managerial ownership (−), government ownership (+), | |
| [46] Khan et al. (2013) | ✓ | | Bangladesh | LT | Firm size (+), media visibility (+), managerial ownership (−), public ownership (+), foreign ownership (+) | |
| [47] Chiu and Wang (2014) | ✓ | | Taiwan | ST | Firm size (+), industry (+), listing in social investment funds (+), impact of global supply chain (+), international capital markets (+) | |

**Table 1.** *Cont.*

| Study | Nature of Study | | Country | Theory | Outcomes | |
|---|---|---|---|---|---|---|
| | Determinants | Motivations | | | Antecedents | Motivations |
| [48] Laidroo and Sokolova (2015) | ✓ | ✓ | Estonia | LT and ST | Firm size (+), firm value (+) shareholder's contribution (+), political perspective (+), legal considerations (+), competitive pressures process standardization (+) | Increased demand of stakeholders' information (+), improved public image (+) |
| [49] Khan et al. (2019) | ✓ | ✓ | Pakistan | N/A | Board gender diversity (+), board education diversity (0), board education background diversity (−), board tenure diversity (+), board age diversity (0), board nationality diversity (+), board ethnicity diversity (0), board size (0), board independence (0), board meeting (0), independent audit committee (+) | Good relations with the labor unions (+); positive firm value (+), and increased accountability (+) |
| [50] Aboud and Diab (2018) | ✓ | ✓ | Egypt | AT | Firm size (0), organizational performance (+), firm value (+), capital expenditure (+), cultural specificity (−), regulatory frameworks (+), shareholders' conflicts (+), management decisions (+), negotiation (+) | Environment friendly engagement (+), financial stability (+) and positive firm value (+) |
| [51] Sun et al. (2018) | ✓ | ✓ | China | ST | Firm size (+), growth rate (0), regulatory pressures (+), stakeholder influence (+) | Recognition of firms' investment capability (+), increased interaction/engagement with the investors (+) |
| [52] Orazalin (2019) | ✓ | ✓ | Kazakhstan | LT | Firm size (+), firm age (+), transparent information (+), interests of depositors and other stakeholders (+), societal pressure (−), independence of board (+) | Uncertainty avoidance (+), increased accountability and responsibility (+) |
| [53] Ramananda and Ataha (2019) | ✓ | ✓ | Indonesia | LT and ST | Firm size (+), firm performance (+), profitability (+), stakeholder's interests (+), sustainability orientation (+), social media consideration (+), positive image (+) | Sustainable community and environmental development (+), proactive in engagement and increased accountability (+) |

**Table 1.** *Cont.*

| Study | Nature of Study | | Country | Theory | Outcomes | |
|---|---|---|---|---|---|---|
| | **Determinants** | **Motivations** | | | **Antecedents** | **Motivations** |
| [54] Khan et al. (2019) | ✓ | ✓ | Pakistan and Turkey | RBV | Firm size (+), age of assets (+), board size (+), managerial ownership (+), legal regulatory guideline (+), environmental concerns (+), management decision making (+) | Sustainable utilisation of resources for environmental development (+), proactive engagement (+) |
| [55] Daas and Alaraj (2019) | ✓ | ✓ | Jordan | LT | Firm size (+), environmental concerns (+), steady growth (+) | Sustainable corporate growth (+), claims of internal initiatives (+) |
| [56] Hamrouni et al. (2019) | ✓ | ✓ | Tunisia | ST and AT | Firm size (0), profitability (+), regulatory pressures (+), eco-friendly practices (+), stakeholder pressure (+) | Recognition of firms' investment capability (+), better management of portfolios (+) |
| [57] Sekhon and Kathuria (2019) | ✓ | ✓ | India | AT, LT and ST | Firm size (+), industry size (+), market regulatory pressure (+), level of competition (+), environmental concerns (+), social responsiveness (+) | Improved brand image and employee morale (+), increasing interest towards social responsibilities (+) |
| [58] Acar and Temiz (2019) | ✓ | ✓ | Turkey | LT and ST | Firm size (0), transparency of information (+), regulatory pressures (+), governmental pressures (+), environmental concerns (+), highly focusing on interests and demands of stakeholders (+) | More visibility of corporate environmental performance (+), enhance motivation to adopt more transparent processes (+) |
| [59] Souror et al. (2020) | ✓ | ✓ | Egypt | LT | Firm size (0), element of independence (+), investment capability (+), management of risk (+), political influence (+), societal expectations (+), regulatory pressure (+) | Fair business practices (+) |
| [60] Bhatia and Makkar (2020) | ✓ | ✓ | India | N/A | Firm size (+), industry (+), income inequality (−), environmental concerns (+), international listing (+), board independence (+) | Corporate accountability (+) |
| [61] Zamir et al. (2020) | ✓ | ✓ | Pakistan | LT | Firm size (+), investment sensitivity (−), firm value (+), regulatory pressure (+), environmental concerns (+), investment efficiency (+) | Corporate investment efficiency (+), positive firm value and increased accountability (+) |

**Table 1.** *Cont.*

| Study | Nature of Study | | Country | Theory | Outcomes | |
|---|---|---|---|---|---|---|
| | Determinants | Motivations | | | Antecedents | Motivations |
| [62] Maama (2020) | ✓ | ✓ | South Africa | IT | Firm size (+), firm value (+), firm age (+), political perspective (+), influence of institutional environment (+) | Influence of governments (+), improved accounting practices (+) |
| [63] Ali and Frynas (2018) | ✓ | | Pakistan | Institutional Theory | CSR Standard Setting Institutions (+), colloboration with NGOs (+), CSR forums and networks (+) | |
| [5] Amran et al. (2014) | ✓ | | Asian-pacific nations | LT & RBV | Board Size (0), board gender diversity (0), board independence (0), organizational CSR related vision and mission (+), CSR committee (+), Collaboration with NGOs (+) | |
| [64] Belal and Cooper (2011) | ✓ | | Bangladesh | PE | Lack of public awareness (−), lack of legal requirements (−), Lack of resources (−), departure from shareholder wealth maximization objective | |
| [65] Belal and Owen (2007) | ✓ | | Bangladesh | N/A | Economically powerful stakeholders (notable parent companies, international buyers, and investors demand) (+), weak institutions is reason of absence of disclosure (−), Enhancement of corporate image (+) | |
| [66] Chapple and Moon (2005) | ✓ | | Malaysia, Indonesia, Philipnines, South Korea, India, Singapore, Thailand | LT, ST | Country (+), internationalization (+), globalization (+) | |
| [67] Choi (1998) | ✓ | | South Korea | N/A | Size (+), industry (+), financial performance (0), auditing (+), Sales growth rate (+) | |

**Table 1.** *Cont.*

| Study | Nature of Study | | Country | Theory | Outcomes | |
|---|---|---|---|---|---|---|
| | Determinants | Motivations | | | Antecedents | Motivations |
| [68] De-Villiers and Johannes (1999) | ✓ | | South Africa | N/A | Absence of legal requirements (−), non-availability of data (−), No motivation for disclosure (−) | |
| [69] De-Villiers and Barnard (2000) | ✓ | | South Africa | LT | Size (+), industry (+), fear of liability (−), listed companies (+) | |
| [70] De-Villiers and Van Staden (2006) | ✓ | | South Africa | LT | Industry (+), non-existance of need to legitimize corporate actions (−) | |
| [71] Garas and ElMassah (2018) | ✓ | | UAE | LT | Firm size (+), assets management (+), managerial ownership (−), market regulatorty pressure (+), societal concerns (+), separation of powers (+), independence of board (+) | |
| [72] Giannarakis (2014) | ✓ | | Greece | N/A | Firm size (0), information flow (−), consumer staple (−), stakeholders interests (+), policy regulators pressures (+) | |
| [73] Gunawan (2007) | ✓ | | India | LT, ST | Stakeholer influence (+), size (+), financial performance (0), firm age (0) | |
| [74] Islam and Deegan (2008) | ✓ | | Bangladesh | LT, ST, IT | Powerful stakeholders (e.g., international buyers, NGOs), demands and global expectations (+) | |
| [75] Issa and Fang (2019) | ✓ | | UAE | ST | Board gender diversity (0), Board independence (0), CEO duality (−), board size (+) | |
| [76] Katmon et al. (2019) | ✓ | | Malaysia | AT and Resource Dependency Theory | board gender diversity (0), board education diversity (+), board education backgrough diversity (0), board tenure diversity (+), board age diversity (−), board nationality diversity (−), board ethnicity diversity (0), board size (0), board independence (+), board meeting (+), independent audit committee (0) | |

**Table 1.** *Cont.*

| Study | Nature of Study | | Country | Theory | Outcomes | |
|---|---|---|---|---|---|---|
| | Determinants | Motivations | | | Antecedents | Motivations |
| [77] Kiliç et al. 2015 | ✓ | | Turkey | LT, ST | Board gender diversity (+), board independence (+), board size (0), ownership diffusion (+), company size (+) | |
| [78] Kolk et al. (2010) | ✓ | | China | LT | Nationality (+) | |
| [79] Kuasirikun (2005) | ✓ | | Thailand | N/A | Latent positive attitudes (+), towards social accounting that may result in CSR disclosure | |
| [80] Kuasirikun and Sherer (2004) | ✓ | | Thailand | LT, ST, IT | Country (+) | |
| [81] Liu and Anbumozhi (2009) | ✓ | | China | ST | Government pressure (+), size (+), financial performance (+), geographical location within country (+) | |
| [82] Matuszak et al. (2019) | ✓ | | Poland | N/A | Firm size (+), board size (+), managerial ownership (+), board leadership (+), legal regulatory guideline (+), public welfare (+), shareholder's interests (+) | |
| [83] Momin and Parker (2013) | ✓ | | Bangladesh | LT, IT | External environment of MNCs (informal norms and beliefs, very low expectations for CSR reporting, lax formal reporting regulation, low level of legal implementation), is a major hurdle for CSR reporting Management culture of parent company (+), and enhance corporate Image (+), are the main reasons for MNCs CSR reporting. | |
| [84] Muttakin et al. 2015 | ✓ | | Bangladesh | AT, LT, ST, and Sinaling Theory | Board gender diversity (−), board independence (+), CEO duality (+), foreign director (+), firm size (+), profitability (+), family ownership (−) | |

**Table 1.** *Cont.*

| Study | Nature of Study | | Country | Theory | Outcomes | |
|---|---|---|---|---|---|---|
| | Determinants | Motivations | | | Antecedents | Motivations |
| [85] Ntim and Soobaroyen 2013 | ✓ | | South Africa | Neo Institutional Theory | Board gender diversity (+), board size (0), independent directors (+), CEO duality (0), government ownership (+), institutional ownership (0), block ownership (−), CSR committee (+) | |
| [86] Oh et al. (2011) | ✓ | | Korea | AT | Institutional ownership (+), and foreign ownership (+) | |
| [87] Rahaman et al. (2004) | ✓ | | Ghana | IT | Institutional pressures from World Bank regulatory requirements (+) | |
| [88] Ratanajongkol et al. (2006) | ✓ | | Thailand | LT, PE | Stakeholder Influence (+), size (0), industry (+) | |
| [89] Zeng et al. (2010) | ✓ | | China | N/A | Size (+), industry (+) | |

LT = Legitimacy Theory; AT = Agency Theory; ST = Stakeholder Theory; IT = Institutional Theory; RBV = Resource-based View Theory; Sigt = Signaling Theory; PE = Political Economy Theory; VDT = Voluntary Disclosure Theory; CT = Communication Theory; Bont = Bonding Theory; Acct = Accountability Theory; MT = Multi-theoretical Lenses; CBF = Cost and Benefit Framework; CMT: Critical Mass Theory; N/A = Not Applied; '+' = Significant positive relationship, '−' = Significant negative relationship.

### 3. Review Results

This section presents findings on the critical trends in empirical research, underpinning theories, antecedents, outcomes, geographical location, and measurement of CSR disclosure research. The studies on the determinants of CSR disclosure and its motivations in developing countries were published in 37 different journals (see Appendix A). *Journal of Business Ethics*, *Accounting, Auditing and Accountability Journal*, *Meditari Accountancy Research*, *Managerial Auditing Journal*, and *Corporate Social Responsibility and Environmental Management* are the leading journals in terms of the number of publications. In total, 46.47% of the studies were published in eight different journals (see Appendix A). The first study on the determinants of CSR reporting in India was published in the *International Journal of Accounting* in 1983 (see Figure 1). The number of publications on CSR disclosure has increased substantially over time since 1983, with 86% of the studies having been conducted in the last 15 years.

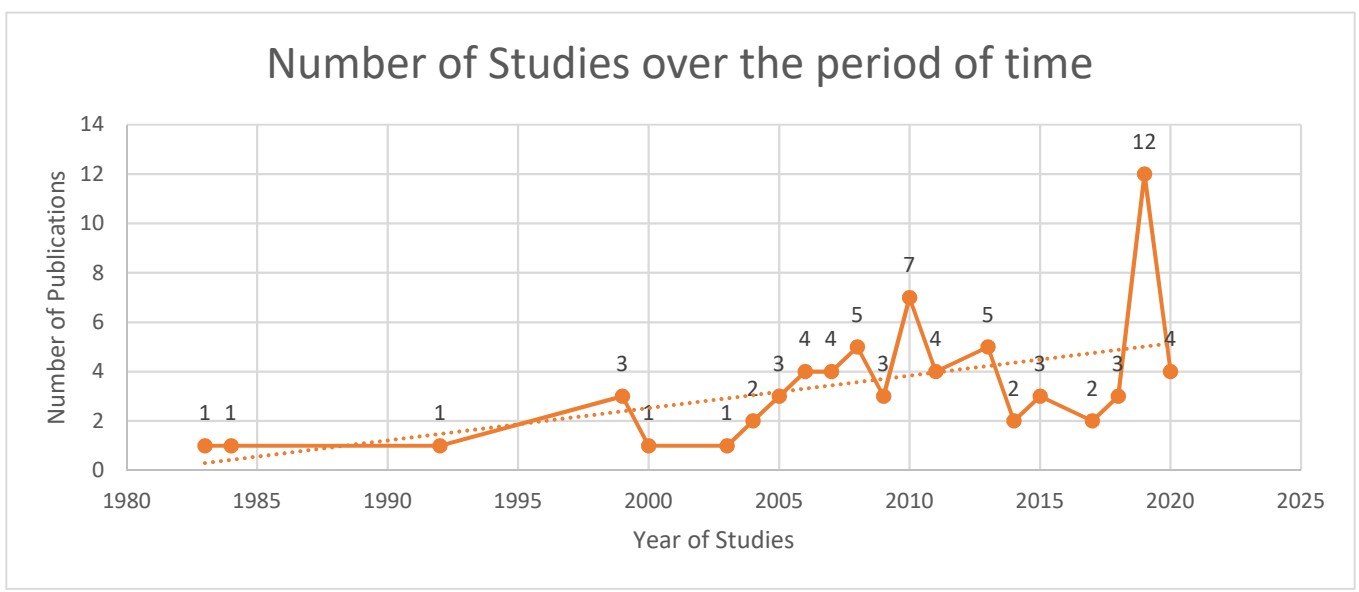

**Figure 1.** Studies on the determinants of CSR disclosure and its motivations over time.

*3.1. Geographical Distribution of CSR Disclosure Studies*

In respect to the geographic distribution of CSR disclosure studies in developing countries, fifty percent of the studies focused on Malaysia, Bangladesh, South Africa, India, and China (see Table 2).

**Table 2.** Geographical distribution of studies on the determinants of CSR disclosure.

| Sr. No | Country | Frequency | %Age |
|--------|---------|-----------|------|
| 1 | Malaysia | 10 | 14.085% |
| 2 | Bangladesh | 8 | 11.268% |
| 3 | South Africa | 7 | 9.859% |
| 4 | India | 5 | 7.042% |
| 5 | China | 5 | 7.042% |
| 6 | Egypt | 3 | 4.225% |
| 7 | Indonesia | 3 | 4.225% |
| 8 | Pakistan | 3 | 4.225% |
| 9 | Thailand | 3 | 4.225% |
| 10 | UAE | 2 | 2.817% |
| 11 | Jordan | 2 | 2.817% |
| 12 | Turkey | 2 | 2.817% |
| 13 | Asian-pacific nations | 2 | 2.817% |

**Table 2.** *Cont.*

| Sr. No | Country | Frequency | %Age |
|---|---|---|---|
| 14 | Taiwan | 2 | 2.817% |
| 15 | Greece | 1 | 1.408% |
| 16 | Poland | 1 | 1.408% |
| 17 | Korea | 1 | 1.408% |
| 18 | Kazakhstan | 1 | 1.408% |
| 19 | Pakistan and Turkey | 1 | 1.408% |
| 20 | Estonia | 1 | 1.408% |
| 21 | Tunisia | 1 | 1.408% |
| 22 | Saudi-Arabia | 1 | 1.408% |
| 23 | Middle-East | 1 | 1.408% |
| 24 | Emerging Countries | 1 | 1.408% |
| 25 | Mauritius | 1 | 1.408% |
| 26 | Ghana | 1 | 1.408% |
| 27 | Malaysia, Indonesia, Philippines, South Korea, India, Singapore, Thailand | 1 | 1.408% |
| 28 | South Korea | 1 | 1.408% |
| | Total | 71 | 100% |

### 3.2. Theoretical Perspectives Used in CSR Disclosure Studies

The theories to explain the determinants of CSR disclosure in developing countries include legitimacy theory, stakeholder theory, agency theory, resource-based view theory, and a combination thereof (see Table 3). Legitimacy theory (22.54%), stakeholder theory (8.45%), and their combination (9.86%) are the most prevalent theoretical frameworks explaining CSR disclosure. Twenty seven percent (26.76%) of the studies used no theory to explain the relationships.

**Table 3.** Theoretical perspectives used in CSR disclosure literature.

| Sr. No | Theory/Theories | Frequency | %Age |
|---|---|---|---|
| 1 | Legitimacy Theory | 16 | 22.54% |
| 2 | Legitimacy Theory and Stakeholder Theory | 7 | 9.86% |
| 3 | Stakeholder Theory | 6 | 8.45% |
| 4 | Institutional Theory | 5 | 7.04% |
| 5 | Agency Theory | 3 | 4.23% |
| 6 | Political Economy Theory | 3 | 4.23% |
| 7 | Agency Theory, Legitimacy Theory and Stakeholder Theory | 3 | 4.23% |
| 8 | Resource Based View Theory | 2 | 2.82% |
| 9 | Miscellaneous theories | 2 | 2.82% |
| 10 | Resource Based View Theory and other theories | 1 | 1.41% |
| 11 | Agency Theory and Stakeholder Theory | 1 | 1.41% |
| 12 | Critical Mass Theory | 1 | 1.41% |
| 13 | Agency Theory and Resource Dependency Theory | 1 | 1.41% |
| 14 | Legitimacy and Institutional Theory | 1 | 1.41% |
| 15 | Not Applied | 19 | 26.76% |
| | Total | 71 | 100% |

### 3.3. CSR Disclosure and Its Dimensions

Forty seven percent (47.41%) of the review studies focused on CSR disclosure generally. In terms of the dimensions of CSR disclosure, twenty five percent (25.00%) emphasised environmental disclosure, thirteen percent (12.93%) emphasized community involvement disclosure and ten percent (10.34%) emphasized human resource disclosure (see Table 4).

**Table 4.** CSR Disclosure and the dimensions studied in the extant literature.

| Sr. No | Disclosure Dimensions | Frequency | Percentage |
|--------|----------------------|-----------|------------|
| 1 | Environmental Disclosure | 29 | 25.00% |
| 2 | Human Resource Disclosure | 12 | 10.34% |
| 3 | Product and Consumer Disclosure | 3 | 2.59% |
| 4 | General Disclosure | 2 | 1.72% |
| 5 | Community Involvement Disclosure | 15 | 12.93% |
| 6 | CSR Disclosure | 55 | 47.41% |
|  | Total | 116 | 100% |

*3.4. Measurement of CSR Disclosure and Its Dimensions*

CSR disclosure studies in developing countries examine the determinants and motivations of CSR disclosure based on assessments of quantity and quality. Here, quantity refers to the extent of social and environmental concern, whereas quality refers to the nature of the information, its accuracy, and the authenticity of the information reported as CSR. In total, 70.42% of the studies considered the extent of the disclosure, while 29.58% considered the quality of the disclosure.

*3.5. Drivers of CSR Disclosure*

The results revealed that a wide range of internal and external factors influence CSR disclosure and its dimensions in developing countries. Tables 5 and 6 present the internal and external contextual factors influencing CSR disclosure. The number of studies reporting the respective factors are listed in parenthesis.

**Table 5.** Internal environment.

| Determinants of CSR/Environmental Disclosure | Sig +ve | Insignificant | Sig −ve | Grand Total |
|---|---|---|---|---|
| **Firms Characteristics** | | | | |
| Firm size | 33 | 1 | 7 | 41 |
| Industry | 18 | 0 | 1 | 19 |
| Financial performance | 11 | 2 | 2 | 15 |
| Firm age | 3 | 1 | 1 | 5 |
| Firm value | 4 | 0 | 0 | 4 |
| Leverage | 1 | 1 | 1 | 3 |
| Transparent information | 3 | 0 | 0 | 3 |
| Audit firm size | 1 | 0 | 1 | 2 |
| Managers/accountants positive attitude | 2 | 0 | 0 | 2 |
| Asset management | 1 | 0 | 0 | 1 |
| Capital expenditure | 1 | 0 | 0 | 1 |
| Employees' information | 1 | 0 | 0 | 1 |
| Fear of liability | 0 | 0 | 1 | 1 |
| Investment capability | 1 | 0 | 0 | 1 |
| Lack of resources | 0 | 0 | 1 | 1 |
| Non availability of data | 0 | 0 | 1 | 1 |
| Non-existence of need to legitimize corporate actions | 0 | 0 | 1 | 1 |
| **Corporate Environmental Policies and Concerns** | | | | |
| Environmental concerns | 7 | 0 | 0 | 7 |
| Environmental performance | 2 | 0 | 0 | 2 |
| Institutional environment | 2 | 0 | 0 | 2 |
| Sustainability orientation | 2 | 0 | 0 | 2 |
| Eco friendly practices | 1 | 0 | 0 | 1 |
| Environmental expenditure | 1 | 0 | 0 | 1 |
| Financing for environmental equipment | 1 | 0 | 0 | 1 |
| GRI adoption | 1 | 0 | 0 | 1 |

**Table 5.** *Cont.*

| Determinants of CSR/Environmental Disclosure | Sig +ve | Insignificant | Sig −ve | Grand Total |
|---|---|---|---|---|
| **Governance Characteristics** | | | | |
| Board size | 6 | 0 | 5 | 11 |
| Board independence | 5 | 1 | 4 | 10 |
| Stakeholders' interest/concern | 10 | 0 | 0 | 10 |
| Board gender diversity | 4 | 1 | 4 | 9 |
| CEO duality | 1 | 2 | 2 | 5 |
| Board age diversity | 0 | 2 | 1 | 3 |
| Board education | 1 | 0 | 1 | 2 |
| Board meetings | 1 | 0 | 1 | 2 |
| CSR committee | 2 | 0 | 0 | 2 |
| Independent audit committee | 1 | 0 | 1 | 2 |
| Long term tenure of directors | 2 | 0 | 0 | 2 |
| Foreign directors on board | 1 | 0 | 0 | 1 |
| Vision and mission | 1 | 0 | 0 | 1 |
| **Owners and Shareholders** | | | | |
| Managerial ownership | 2 | 3 | 1 | 6 |
| Disperse ownership | 3 | 0 | 1 | 4 |
| Foreign ownership | 3 | 0 | 1 | 4 |
| Government ownership | 3 | 1 | 0 | 4 |
| Institutional ownership | 2 | 1 | 1 | 4 |
| Shareholder contribution | 3 | 1 | 0 | 4 |
| Public ownership | 1 | 0 | 0 | 1 |

Sig +ve: Significant positive relationship; Sig −ve: Significant negative relationship; Insig: Insignificant.

**Table 6.** General external environment.

| Determinants of CSR /Environmental Disclosure | Sig +ve | Insignificant | Sig −ve | Grand Total |
|---|---|---|---|---|
| **Political and Legal Factors** | | | | |
| Regulatory pressure | 15 | 0 | 0 | 15 |
| Absence of legal requirements | 0 | 2 | 3 | 5 |
| Political development/pressure | 3 | 0 | 0 | 3 |
| Government pressure | 2 | 0 | 0 | 2 |
| Dependence on government | 1 | 0 | 0 | 1 |
| Low level of legal implementation | 0 | 0 | 1 | 1 |
| Media visibility/pressure | 1 | 0 | 0 | 1 |
| Political system | 1 | 0 | 0 | 1 |
| Weak institutions of the country | 0 | 0 | 1 | 1 |
| **Global Issue** | | | | |
| International buyer pressure | 2 | 0 | 0 | 2 |
| Global supply chain | 1 | 0 | 0 | 1 |
| Globalization | 1 | 0 | 0 | 1 |
| Pressure from regulatory bodies e.g., World Bank | 1 | 0 | 0 | 1 |
| International NGOs | 1 | 0 | 0 | 1 |
| **Normative Institution (CSR Promoting Institutions)** | | | | |
| Collaboration with NGOs | 2 | 0 | 0 | 2 |
| CSR forums and networks | 1 | 0 | 0 | 1 |
| CSR standard setting institutions | 1 | 0 | 0 | 1 |

**Table 6.** *Cont.*

| Determinants of CSR /Environmental Disclosure | Sig +ve | Insignificant | Sig −ve | Grand Total |
|---|---|---|---|---|
| **Social Cultural Factors** | | | | |
| Country specific factors | 4 | 0 | 0 | 4 |
| Public pressure | 2 | 2 | 0 | 4 |
| Cultural factor | 2 | 0 | 0 | 2 |
| Cultural specificity | 0 | 1 | 0 | 1 |
| Income inequality | 0 | 1 | 0 | 1 |
| Lack of public awareness | 0 | 0 | 1 | 1 |
| Public welfare | 1 | 0 | 0 | 1 |
| Social media concerns | 1 | 0 | 0 | 1 |
| Low public expectations for CSR reporting | 0 | 0 | 1 | 1 |
| **Industry Level Factors** | | | | |
| Level of competition | 2 | 0 | 0 | 2 |
| Capital market | 1 | 0 | 0 | 1 |
| Customer concerns | 1 | 0 | 0 | 1 |
| Market forces | 1 | 0 | 0 | 1 |
| Multiple listing | 1 | 0 | 0 | 1 |
| Overseas listing | 1 | 0 | 0 | 1 |
| Stock market listing | 1 | 0 | 0 | 1 |
| Suppliers | 0 | 1 | 0 | 1 |
| Systematic risk | 1 | 0 | 0 | 1 |

Sig +ve: Significant positive relationship; Sig −ve: Significant negative relationship; Insig: Insignificant.

### 3.5.1. Internal Factors

Internal factors include company characteristics, corporate environmental policies and concerns, corporate governance, and owners and shareholders. Concerning company characteristics, firm size (33), industry (19), and financial performance (11) are the predominant factors driving CSR disclosure in developing countries. In addition to this, the company characteristics of the firm's value (4), transparent information (3), the firm's age (3), leverage (1), the firm's audit size (1), government dependency (1), and capital expenditure (1) also influence the disclosure of CSR information. These results are consistent with earlier reviews, notably [3,4]. In addition to the above, the lack of resources (1), fear of liability (1), non-availability of CSR-related data (1), and non-existence of a need to legitimize corporate actions (1) are reasons for the lack of disclosure of CSR-related information in developing countries. Furthermore, the level of information transparency (3), the positive attitude of managers and accountants (2), the firm's investment capability (1), and asset management positively contributed to the disclosure of CSR information. Corporate environmental policies and concerns also influence disclosure (see Table 5). The results show that environmental concerns (7), sustainability orientation (2), ecofriendly practices (1) and GRI adoption (1) positively influence level of CSR disclosure. Additionally, the factors of environmental performance (2), institutional environment (2), environmental expenditures (1), and financing for environmental equipment (1) correspond with disclosure. Corporate governance characteristics such as the board size, an independent audit committee, board independence, board meetings and board diversity— including gender, qualification, tenure, and age—appear to have mixed influence on the disclosure of CSR information (see Table 5). However, the presence of a CSR committee (2), foreign directors on the board (1), and a vision and mission (1) positively influence CSR disclosure. Shareholders' concerns and ownership types also contribute to CSR disclosure. Contrary to other types of ownership, government ownership, foreign ownership, and institutional ownership are associated with CSR reporting as well.

### 3.5.2. External Environment

External environmental factors play a substantial role in the promotion of CSR disclosure in developing countries. The results showed that despite poor law enforcement in developing countries, regulatory pressures positively influence CSR disclosure (see Table 6). Similar to the above, studies have found that the absence of legal requirements, weak institutions, and a low level of implementation are negatively related to social and environmental disclosure. In addition, government pressure and media pressure positively influence CSR disclosure. Social–cultural factors and global supply chains also positively influence CSR disclosure. Global stakeholders such as international buyers, global supply chains, globalization, international NGOs, and international regulatory bodies—e.g., The World Bank—positively influence social and environmental disclosure in developing countries. Normative institutions such as companies' collaboration with NGOs, CSR forums and networks, and CSR standard setting institutions in developing countries were found to have a significant positive relationship with CSR disclosure. However, a lack of public concern about social and environmental issues negatively influences social and environmental disclosure.

The industry-level factors that positively influence CSR disclosure in developing countries include customers' concerns, the level of competition, stock market listing, and overseas listing. Financial market forces also appear to play a positive role in the promotion of CSR reporting. Suppliers' concerns negatively influence CSR disclosure.

### 3.6. Motivations for CSR Disclosure

The main motivations driving companies in developing countries to disclose CSR information are to gain corporate reputation and to enhance financial performance. Other reasons noted in the literature are to demonstrate corporate accountability, to manage key stakeholders, and to attract investment opportunities (see Table 7).

**Table 7.** Motivations of CSR disclosure.

| Motivations of CSR Disclosure | F | %Age |
|---|---|---|
| **Corporate Accountability** | | |
| Demonstrate corporate accountability | 4 | 5.56% |
| Improve accounting practices | 1 | 1.39% |
| Total | 5 | |
| **Corporate Reputation** | | |
| Showcase corporate environmental performance | 3 | 4.17% |
| Improve public image | 2 | 2.78% |
| Exhibit environment friendly engagement | 1 | 1.39% |
| Improve environmental management systems | 1 | 1.39% |
| Promote fair business practices | 1 | 1.39% |
| Total | 8 | |
| **Financial Performance** | | |
| Drive corporate performance | 1 | 1.39% |
| Promote positive firm value | 3 | 4.17% |
| Contribute to sustainable corporate growth | 1 | 1.39% |
| Improve financial stability | 1 | 1.39% |
| Eliminate uncertainty in reporting practices | 1 | 1.39% |
| Enhance corporate profitability and social responsibility | 1 | 1.39% |
| Total | 8 | |
| **Investment Opportunities** | | |
| Recognize firms' investment potential | 2 | 2.78% |
| Secure more opportunities for institutional investments | 1 | 1.39% |
| Demonstrate corporate investment efficiency | 1 | 1.39% |
| Total | 4 | |

**Table 7.** *Cont.*

| Motivations of CSR Disclosure | F | %Age |
|---|---|---|
| **Management of key Stakeholders** | | |
| Increase interaction/engagement with investors/ stakeholders | 4 | 5.56% |
| Demonstrate good relations with the labor unions | 1 | 1.39% |
| Respond to increased stakeholders' demand for information | 1 | 1.39% |
| Influence on governments | 1 | 1.39% |
| Reduce political costs | 1 | 1.39% |
| Improve employee morale | 1 | 1.39% |
| Align firm's operations with stakeholders | 1 | 1.39% |
| Total | 10 | |

## 4. Discussion

Our research shows that both external and internal factors drive the social and environmental reporting agenda in developing countries. The key factors identified in the literature as motivating social and environmental disclosure in developing countries include political and legal factors, as well as social and cultural factors such as the public awareness of social issues and media pressure. These findings support prior research stating that CSR is a socially constructed and dynamic concept that is influenced by national contextual (e.g., social, political and cultural) factors [3,90–92]. The same corporate behavior that is acceptable in one region may not be acceptable in another, resulting in varying types of CSR disclosure. In developing countries, as in developed countries, government initiatives (or regulations) and stakeholder expectations are major drivers of CSR reporting. Conversely, the lack of CSR reporting expectations by the government and stakeholders is seen as a major reason for non-disclosure in developing nations. At a firm level, CSR implementation is hindered by managers' perceptions of disclosure not yielding the desired benefits, the high cost of CSR, the time requirement, and a lack of knowledge [93].

Unlike developed countries, firms in the developing world feel little public pressure domestically to practice CSR [3]. CSR disclosure, however, is influenced by global stakeholders such as international buyers, global supply chains, globalization, international NGOs, and international regulatory bodies such as The World Bank.

Normative institutions such as companies' collaboration with NGOs, CSR forums and networks, and CSR standard-setting institutions in developing countries were found to have a significant positive relationship with CSR disclosure. Institutional theory can provide a plausible explanation for this result. According to institutional theory, CSR frameworks and networks, NGOs, and CSR standard-setting institutions are normative institutions that set the appropriate standards for a firm [94,95]. Companies that interact with or are members of CSR-promoting institutions are said to be more aware of CSR issues and are more likely to act in a socially responsible manner [63,90,96]. Based on the significant relationship between CSR-promoting institutions and CSR disclosure, policies to encourage the creation and promotion of such institutions may be needed in order to supplement state institutions in developing countries.

Internal factors which were found to influence CSR disclosure in developing countries include company characteristics, corporate environmental policies and concerns, corporate governance, and owners and shareholders. Company characteristics driving CSR disclosure include: firm size, financial performance, and industry sensitivity. A large size, profits, and environmental sensitivity of operations influence a company's public or social visibility [63]. Various stakeholders—including the media, non-governmental organizations, and the government—may put pressure on a highly visible company to act in a socially and environmentally responsible manner [90,97]. According to research, a socially visible company discloses CSR information in order to be recognized as a legitimate company by reflecting consistency between corporate actions and the practices institutionalized in the environment in which the firm operates [98,99]. In addition to this, company characteristics such as firm value, the firm age, and firm size and leverage also influence the disclosure

of CSR information. The results suggest that older firms, larger firms and firms with a high book value are more likely than newer firms, smaller firms and firms with lower book values to disclose CSR information. Larger firms and firms with a high book value often face greater scrutiny from the public, media, and government and as a result are more likely to disclose their CSR activities. Firms are less likely to disclose CSR information if they are highly leveraged, lack resources, lack CSR related data, fear liability, or do not perceive a need to legitimize corporate actions [3].

Corporate environmental policies and concerns also influence corporate social disclosure. The adoption of social and environmental policies by corporations can be attributed to various isomorphic pressures, such as coercive, normative, and mimetic pressures emanating from various regulatory, normative, or cognitive institutions operating in the institutional environment in which the corporation operates [100].

In reference to corporate governance characteristics, the research showed a significant positive and negative relationship between board size and CSR disclosure. According to Siregar and Bachtiar [101], larger boards are more effective at monitoring activities, but overly large boards lose this effectiveness. The results suggest a mixed relationship between board diversity and CSR disclosure. The proponents of board diversity argue that board diversity with respect to gender, ethnicity, education, and cultural background increases board independence because directors from diverse backgrounds ask questions that homogeneous directors may not [102,103]. As a result, having board diversity boosts the board's independence. Independence has been argued to be one of the most important factors influencing entities' accountability and disclosure practices [103].

Shareholders' concerns and ownership types also contribute to CSR disclosure. Contrary to other types of ownership, government ownership, foreign ownership, and institutional ownership positively influence CSR reporting, as well. This result can be attributed to coercive pressure from powerful stakeholders [31,87], including the government, foreign owners, and institutional investors. However, on some occasions, dispersed ownership, foreign ownership, and institutional ownership showed a negative relationship with social and environmental disclosure. This negative relationship can be explained by the small percentage of dispersed, foreign, and institutional ownership in the focal companies.

## 5. Conclusions

The main aim of this research was to systematically analyse and synthesise the empirical literature on the drivers and motivations of CSR disclosure in developing countries. In accomplishing this aim, we surveyed 71 empirical studies focusing on the determinants and motivations of CSR disclosure in developing countries. The results revealed that both internal factors and external factors influence the disclosure of CSR information. In the internal category, social and environmental disclosure are influenced by company characteristics such as size, industry, financial performance, corporate governance elements such as board size and board independence, and types of ownership, particularly foreign, government and institutional ownership. In addition, corporate polices and concerns also influence the disclosure of CSR-related information. In the external category, political and legal factors such as regulatory pressures, government pressure, media concerns, industry-level factors such as the level of industry competition, customers' concerns, the multiple listing of a firm, and social–cultural factors positively influence social and environmental reporting. Furthermore, global value chains, international buyers, international NGOs, and international regulatory bodies pressure companies in developing countries to disclose social and environmental information Between the two, external factors have a stronger influence on social and environmental reporting in developing countries compared to internal factors. This finding is consistent with the existing review studies of CSR disclosure in developed and developing countries [3,104]. With regard to the motivations of CSR disclosure, companies appear to disclose social and environmental information in order to gain corporate reputation, enhance financial performance, secure investment, and manage key stakeholders.

Our study is not free from limitations. This review covered articles published in the English language and ABS-ranked 2018 journals, and excluded books, book chapters, conference proceedings, and work published in predatory journals. Furthermore, this review considered empirical research papers only, and may have ignored the antecedents and motivations of social and environmental disclosure where data is unavailable or scarce. The noticeable lack of relevant studies on the determinants and outcomes of CSR disclosure in low–middle income countries, especially outside the Anglophone world, may miss some aspects of social and environmental disclosure in the current review.

Based on our review, we note several gaps in the literature which are worthy of future research. Prior research has paid considerable attention to the determinants, the theoretical perspectives used, and the measurement of social and environmental disclosure in developing countries. Future research should focus on the determination of the authenticity, accuracy, and reliability of disclosure studies by employing verifiable methods. Furthermore, many studies focused on the determination of the quantity or quality of CSR disclosure in developing countries but fail to assess the comprehensiveness of social and environmental reporting. The extant studies also neglect to consider differences in determinants and motivators by region. Additionally, the extent reviews have focused on large public traded companies. Few studies have emphasized small–medium enterprises (SMEs). Future research should examine prevailing SMEs' social and environmental issues, and the measures taken to address them.

While studies report that entrepreneurship can benefit the economy by creating wealth and jobs, and competition [105–110], a recent study by Zhang and colleagues [111] found that an organization's entrepreneurial orientation has an effect on its CSR activities. The study found that CEOs with an entrepreneurial mindset are more likely to engage their companies in CSR innovation, rather than corporate philanthropy. CSR innovation focuses on expanding core businesses or developing new forms of business to address social and environmental issues [112], whereas corporate philanthropy refers to the financial contributions made by businesses to society and charity [113–115]. Corporate entrepreneurialism appears to contribute to CSR activities and may serve as a predictor of CSR disclosure in developing countries. Surprisingly, none of the 71 articles analysed in this study discussed the role of entrepreneurship as a catalyst for CSR disclosure. As a result, future research should examine the role of entrepreneurship and innovation as a contributor to CSR disclosure.

**Author Contributions:** Conceptualization, W.A. and J.W.; methodology, M.H.; data curation, W.A. and M.H.; writing—original draft preparation, W.A. and J.W.; writing—review and editing, J.W. and W.A. All authors have read and agreed to the published version of the manuscript.

**Funding:** This research received no external funding.

**Institutional Review Board Statement:** Not applicable.

**Informed Consent Statement:** Not applicable.

**Data Availability Statement:** Not applicable.

**Conflicts of Interest:** The authors declare no conflict of interest.

**Appendix A**

**Table A1.** Journal—Publications on the Determinants and Motivations of CSR Disclosure.

| Sr. # | Journal | Frequency | %Age |
|-------|---------|-----------|------|
| 1 | Journal of Business Ethics | 7 | 10% |
| 2 | Accounting, Auditing & Accountability Journal | 4 | 6% |
| 3 | Meditari Accountancy Research | 4 | 6% |
| 4 | Managerial Auditing Journal | 4 | 6% |
| 5 | Social Responsibility Journal | 4 | 6% |
| 6 | Corporate Social Responsibility and Environmental Management | 4 | 6% |

**Table A1.** *Cont.*

| Sr. # | Journal | Frequency | %Age |
|---|---|---|---|
| 7 | International Journal of Islamic and Middle Eastern Finance and Management | 3 | 4% |
| 8 | Critical Perspectives on Accounting | 3 | 4% |
| 9 | Journal of Accounting in Emerging Economies | 2 | 3% |
| 10 | International Journal of Law and Management | 2 | 3% |
| 11 | The International Journal of Business in Society | 2 | 3% |
| 12 | Management Decision | 2 | 3% |
| 13 | Asian Review of Accounting | 2 | 3% |
| 14 | Accounting, Organizations and society | 2 | 3% |
| 15 | Pacific Accounting Review | 2 | 3% |
| 16 | Business Strategy and the Environment | 2 | 3% |
| 17 | Journal of Cleaner Production | 2 | 3% |
| 18 | International Journal of Law and Management | 1 | 1% |
| 19 | Critical Perspectives on International Business | 1 | 1% |
| 20 | Journal of Applied Accounting Research | 1 | 1% |
| 21 | Management Research Review | 1 | 1% |
| 22 | International Journal of Managerial Finance | 1 | 1% |
| 23 | Baltic Journal of Management | 1 | 1% |
| 24 | International Journal of Emerging Markets | 1 | 1% |
| 25 | International Journal of Accounting | 1 | 1% |
| 26 | The International Journal of Accounting | 1 | 1% |
| 27 | Journal of Accounting and Public Policy | 1 | 1% |
| 28 | Management & Accounting Review | 1 | 1% |
| 29 | Advances in International Accounting | 1 | 1% |
| 30 | Issues in Social and Environmental Accounting | 1 | 1% |
| 31 | The British Accounting Review | 1 | 1% |
| 32 | Business & Society | 1 | 1% |
| 33 | Qualitative Research in Accounting & Management | 1 | 1% |
| 34 | Corporate Governance | 1 | 1% |
| 35 | Corporate governance: an International Review | 1 | 1% |
| 36 | Gender in Management: An International Journal | 1 | 1% |
| 37 | Journal of Corporate Finance | 1 | 1% |
| | Total | 71 | 100% |

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
