# Peer review of "Determinants/Motivations of Corporate Social Responsibility Disclosure in Developing Economies: A Survey of the Extant Literature"

_sustainability, doi:10.3390/su14063474_

Round 1

Reviewer 1 Report

The subject of the article is extremely important, especially nowadays, when the approach to sustainable development, including social responsibility, is widely promoted and implemented. 

However, the wording of the title itself is somewhat misleading, by no means from my perspective. You use the wording in developing countries, and the work actually highlights the approaches represented by specific countries: Malaysia, India, Egypt, Indonesia, Portugal, South Africa, and Bangladesh. Quite late in the text itself it was explained why the choice fell on these particular countries - I would suggest adding a mention of it in the abstract or in the introduction. Or to clarify (slightly change) the title of the article.

I appreciate the enormous amount of work that the authors put into conducting a comprehensive literature review. I just wonder if the existence of the identified dependencies in social enterprises should not be verified responsibly, even on a small sample - it would increase the utilitarian nature of the study.

In my opinion, both the introductory and the concluding parts were prepared quite "sparingly". The theory of corporate social responsibility, its genesis and evolution over the years provides a solid basis for supplementing the initial arguments. In case of termination, I miss a discussion thread. It should be completed.

I wish the author's success in correcting the article.

Good luck and be healthy!

Author Response

Thanks for your constructive comments and suggestions, which have helped us to develop the paper further. We have taken all your comments and suggestions on board. We reproduce each of your original comments in italics followed in turn by our responses.

Reviewer 1 Comments

  1. The subject of the article is extremely important, especially nowadays, when the approach to sustainable development, including social responsibility, is widely promoted and implemented. 

Thanks for favorable comments and we appreciate your valuable comments.

  1. However, the wording of the title itself is somewhat misleading, by no means from my perspective. You use the wording in developing countries, and the work actually highlights the approaches represented by specific countries: Malaysia, India, Egypt, Indonesia, Portugal, South Africa, and Bangladesh. Quite late in the text itself it was explained why the choice fell on these particular countries - I would suggest adding a mention of it in the abstract or in the introduction. Or to clarify (slightly change) the title of the article.

The title to article has been changed to appropriately reflect the contents of the article. The new title is “Determinants /Motivations of Corporate Social Responsibility Disclosure in Developing Countries: A Survey of Extant Literature” (see line 2-3)

  1. I appreciate the enormous amount of work that the authors put into conducting a comprehensive literature review. I just wonder if the existence of the identified dependencies in social enterprises should not be verified responsibly, even on a small sample - it would increase the utilitarian nature of the study.

Thank you once more for your kind words. The purpose of this study is to systematically summarize the existing literature on the determinants of CSR disclosure in developing countries and to highlight the factors that influence CSR disclosure in developing countries. Furthermore, rather than using empirical data from a large sample, this study incorporates the findings of previous studies on the determinants of CSR disclosure into this study to summarize the literature. Therefore, the results cannot be verified/tested on the small sample.

  1. In my opinion, both the introductory and the concluding parts were prepared quite "sparingly". The theory of corporate social responsibility, its genesis and evolution over the years provides a solid basis for supplementing the initial arguments. In case of termination, I miss a discussion thread. It should be completed.

The article's introduction and conclusion have been updated [see lines 37-64; 289-292]. Since the purpose of this research is to examine the determinants/ motivations CSR disclosure in developing countries based on extant published literature. Therefore, the authors feel that “theory of corporate social responsibility, its genesis and evolution over the years” beyond the scope of this study. Therefore, “theory of corporate social responsibility, its genesis and evolution over the years” has not been discussed in this study. However, discussion section has been added into the paper [see lines 215-277].

Looking forward for positive reply.

Kind Regards,

Waris Ali, Jeffery Wilson, and Muhammad Husnain

Reviewer 2 Report

The topic of the article seems to be interesting and up-to-date, important from a practical point of view. However, below I present my comments and suggestions, hoping that my help in improving the article will be taken into account by the authors.

Title

I argue that the title that reads "Corporate Social Responsibility Disclosure in Developing Countries: A Survey of Extant Literature" is not good. There is no problem in it.

Abstract

The abstract does not outline the problem and - importantly - the research goal is missing. Without indicating the purpose of the research, it cannot be verified whether it was achieved by the authors. So - what is the purpose of the research? Why did the authors write this article? What did they want to show / prove? The analysis and synthesis of literature absolutely cannot be considered a research goal.

Introduction

I disagree with the authors' statement that there is no research on CSR in developing countries. I also do not agree that countries such as Portugal, Estonia, Hong Kong, Italy and Sweden should be considered developing. I am based here on the classification of the International Monetary Fund. Once again, I will return to the question about the purpose of the research - it also makes no sense here. So I still don't know what the purpose of this research is. This is a serious mistake. I would like to add that any possible answers to research questions cannot be considered the purpose of the research either. Moreover, there is no research hypothesis, which makes it difficult to determine what the authors' intentions are.

Methodology

The methodology is poorly described. There is no basis for a theoretical methodology. It is not known why the authors chose the multi-step strategy of Denyer and Trandfield - they did not justify it. I consider it unnecessary to repeat the research questions. It would be much better to include the research algorithm here. You could also discuss research questions here, clarify what is going on, because they are not entirely clear. I consider it a mistake to exclude books, book chapters, conference materials, and papers published in predatory journals from the research. Thus, many works were omitted. Proposes that authors review the Sustainability journal for work on CSR. Apart from compilation and presentation of data, no research method was used. It is not enough. Such presentation of the data obtained from the literature review brings nothing new to science.

Discussion

The authors did not hold any discussion.

Summary  - The idea for the article may be good, but the article has many serious drawbacks. I believe that in this form it is not ready for publication, and the importance of the amendments determines its rejection. 

Author Response

Thanks for your constructive comments and suggestions, which have helped us to develop the paper further. We have taken all your comments and suggestions on board. Where we were unable to address the comments, we have provided justification for not incorporating the comments. We reproduce each of your original comments in italics followed in turn by our responses.

  1. The topic of the article seems to be interesting and up-to-date, important from a practical point of view. However, below I present my comments and suggestions, hoping that my help in improving the article will be taken into account by the authors.

Thanks for favorable comments and we appreciate your valuable comments.

Title

  1. I argue that the title that reads "Corporate Social Responsibility Disclosure in Developing Countries: A Survey of Extant Literature" is not good. There is no problem in it.

The title of the article has been changed to appropriately reflect the contents of the article. The new title is “Determinants / Motivations of Corporate Social Responsibility Disclosure in Developing Countries: A Survey of Extant Literature” [see line 2-3]

Abstract

  1. The abstract does not outline the problem and - importantly - the research goal is missing. Without indicating the purpose of the research, it cannot be verified whether it was achieved by the authors. So - what is the purpose of the research? Why did the authors write this article? What did they want to show / prove? The analysis and synthesis of literature absolutely cannot be considered a research goal.

The purpose of the article has been mentioned in the abstract [see line 9-29].

Introduction

  1. I disagree with the authors' statement that there is no research on CSR in developing countries. I also do not agree that countries such as Portugal, Estonia, Hong Kong, Italy and Sweden should be considered developing. I am based here on the classification of the International Monetary Fund. Once again, I will return to the question about the purpose of the research - it also makes no sense here. So I still don't know what the purpose of this research is. This is a serious mistake. I would like to add that any possible answers to research questions cannot be considered the purpose of the research either. Moreover, there is no research hypothesis, which makes it difficult to determine what the authors' intentions are.

The researchers have updated the introduction section, which also include the research's purpose [see lines 36-51]. Furthermore, in accordance with the reviewers' recommendations, the authors excluded studies aimed at Portugal, Hong Kong, Italy, and Sweden from the study and included more studies examining the determinants of social and environmental disclosure in developing countries [see Table 1]. The authors have now examined 71 empirical studies from developing countries. Since the purpose of this research is to examine the determinants/ motivations CSR disclosure in developing countries based on extant published literature. Therefore, the authors feel that “there is no research hypothesis….” beyond the scope of this study. Therefore, hypotheses have not been developed in this study.

Methodology

  1. The methodology is poorly described. There is no basis for a theoretical methodology. It is not known why the authors chose the multi-step strategy of Denyer and Trandfield - they did not justify it. I consider it unnecessary to repeat the research questions. It would be much better to include the research algorithm here. You could also discuss research questions here, clarify what is going on, because they are not entirely clear. I consider it a mistake to exclude books, book chapters, conference materials, and papers published in predatory journals from the research. Thus, many works were omitted. Proposes that authors review the Sustainability journal for work on CSR. Apart from compilation and presentation of data, no research method was used. It is not enough. Such presentation of the data obtained from the literature review brings nothing new to science.

The authors have used Denyer and Trandfield's multi-step strategy to search for articles to be studied in this research. In fact, authors have used the systematic approach to select the articles studied in this research. Further, the repetition of research questions has been removed from the methodology section.

Discussion

  1. The authors did not hold any discussion.

The discussion section has been added into the article [see lines 2014-277].

Looking forward for positive reply.

Kind Regards,

Waris Ali, Jeffery Wilson, and Muhammad Husnain

Reviewer 3 Report

In the intention of the authors, the review manuscript was a systematic review of the drivers and motivations of corporate social responsibility (CSR) disclosure in developing countries based on a survey of 51 empirical studies. The added value of the paper is that it focuses on developing countries that are still not well understood in the literature. However, I believe that the article does not fully meet the conditions and assumptions for systematic review, especially in terms of the presentation of results.

Furthermore, the manuscript does not meet the editorial requirements of the “Sustainability” journal for text formatting.

In the following, detailed comments are presented in the order of their formulation.

For what purpose do we ask the same research questions twice? (lines 50-56) and (71-75).

  • Table 1 needs to be sorted out! It is too long.
  • This table should not be referred to individual articles, the more the column ‘Antecedents’ contains the same terms repeatedly, but rather shows the frequency of occurrence of a particular issue/problem in the articles studied.
  • In my opinion, Table 2 does not contribute much; it could be transferred to the appendix; furthermore, this table does not answer any of the research questions, but is only a representation of the methodological background..
  • The information in Tables 2-6 could not be combined into a table, which would save a lot of space?
  • The sections: ‘Measurement of CSR Disclosure and Its Dimensions’ and ‘Drivers of CSR Disclosure’ require a definite extension. There is a lack of information on what quantitative and qualitative methods were used, what detailed research methods were carried out (surveys, interviews, etc.). etc.), how the variables were operationalized, how many and which entities were tested…
  • The paper shows the direction of the influence of particular drivers (using ‘Sig + ve or -ve’ in Tables 7 and 8) but it is a question of which method has achieved the above results.
  • I also did not find an attempt to explain the influence of individual factors, This is more important because Tables 7 and 8 show that the influence of certain factors was different ( in some studies, positive, in others, negative). This definitely requires a comment!
  • In summary, there is no clear emphasis on how the results for developing countries differ from those for developed countries, which was probably the main intention of the authors.

Summing up, I have the impression that the authors have not read the articles fully, but have only been limited to a brief look, which is not acceptable in the case of a systematic review.

Author Response

Thanks for your constructive comments and suggestions, which have helped us to develop the paper further. We have taken all your comments and suggestions on board. Where we were unable to address the comments, we have provided justification for not incorporating the comments. We reproduce each of your original comments in italics followed in turn by our responses.

  1. In the intention of the authors, the review manuscript was a systematic review of the drivers and motivations of corporate social responsibility (CSR) disclosure in developing countries based on a survey of 51 empirical studies. The added value of the paper is that it focuses on developing countries that are still not well understood in the literature. However, I believe that the article does not fully meet the conditions and assumptions for systematic review, especially in terms of the presentation of results.

Furthermore, the manuscript does not meet the editorial requirements of the “Sustainability” journal for text formatting.

In the following, detailed comments are presented in the order of their formulation.

For what purpose do we ask the same research questions twice? (lines 50-56) and (71-75).

The repetition of research questions has been addressed by removing the research questions from the section 2.1

  1. Table 1 needs to be sorted out! It is too long.
  • This table should not be referred to individual articles, the more the column ‘Antecedents’ contains the same terms repeatedly, but rather shows the frequency of occurrence of a particular issue/problem in the articles studied.

Since the Table 1 contains the details of the study, nature of relationship, and context of the study. Therefore, it is not possible to shorten the length of this table. However, the frequency of the individual result has been mentioned in the analysis section of the article.

  1. In my opinion, Table 2 does not contribute much; it could be transferred to the appendix; furthermore, this table does not answer any of the research questions, but is only a representation of the methodological background.

Table 2 has been mentioned in the Appendix as Appendix A [see line 534].

  1. The information in Tables 2-6 could not be combined into a table, which would save a lot of space?

Table 2-6 contain different kinds of information; therefore, Table 2-6 could not be combined.

  1. The sections: ‘Measurement of CSR Disclosure and Its Dimensions’ and ‘Drivers of CSR Disclosure’ require a definite extension. There is a lack of information on what quantitative and qualitative methods were used, what detailed research methods were carried out (surveys, interviews, etc.). etc.), how the variables were operationalized, how many and which entities were tested…

The explanation for the measurement for CSR disclosure and its dimensions have been provided [see lines 157-160]. Further, the section on drivers of CSR disclosure has been extended [see lines 166-206]. Table 4 summarizes the focus of existing research. Whether the existing research investigated CSR disclosure, its dimensions, or both, while Table 5 focuses on the measurement of of CSR disclosure or its dimensions. To report that how the variables were operationalized by the existing studies is not possible at this stage of research, therefore, this point could not be addressed.

  1. The paper shows the direction of the influence of particular drivers (using ‘Sig + ve or -ve’ in Tables 7 and 8) but it is a question of which method has achieved the above results.

Sig +Ve or -ve reported in the Table 7 and 8 are the directions of relationships reported by the existing scholarships

  1. I also did not find an attempt to explain the influence of individual factors, This is more important because Tables 7 and 8 show that the influence of certain factors was different ( in some studies, positive, in others, negative). This definitely requires a comment!

This point has been addressed by highlighting the influence of individual factors in the analysis section [see lines 166-206]. Further, the justification about nature of relationship has been mentioned in the discussion section [see lines 214-276].

  1. In summary, there is no clear emphasis on how the results for developing countries differ from those for developed countries, which was probably the main intention of the authors.

This was not the main purpose of this research. The main purpose of the research was to examine the determinants of CSR disclosure in developing countries. However, how the results in developing are different form developed countries are reported in the discussion section [see lines 214-276].

Looking forward for positive reply.

Kind Regards,

Waris Ali, Jeffery Wilson, and Muhammad Husnain

Reviewer 4 Report

Dear author(s),

I think the logical chain of the paper itself is clear. However as a paper to be published in the JCR journal, it seems that the article lacks some depth, so I suggest that the author make the following changes.

First of all, what is the significance of the article making such a statistical analysis? Is it to highlight policy recommendations for economic development in developing countries? We have seen a lot of detailed statistics, but we have not seen this part of the discussion, so I think the author needs to add the relevant part of the theoretical discussion.

I would like to suggest the following changes. Economic theories show that entrepreneurs are the driving force of the market economy, not the government. The authors should emphasize how the entrepreneurs play their role in the market economy and how this could affect CSR.

I recommend the authors to refer to the following books and articles and provide a structural section on entrepreneurship, economic development CSR. Of course, apart from the following references, the authors can also address other theoretical works that they know and that is related to the above topics.

Huerta de Soto, J. (2010). Socialism, Economic Calculation and Entrepreneurship. Edward Elgar.

Foss, N. J., & Klein, P. G. (2012). Organizing entrepreneurial judgment: A new approach to the firm. Cambridge University Press.

Harper, D. A. (2003). Foundations of entrepreneurship and economic development. Routledge.

Espinosa, V. I., Wang, W. H., & Zhu, H. (2020). Israel Kirzner on dynamic efficiency and economic development. Procesos de mercado: revista europea de economía política17(2), 283-310.

Another question is, why choose these developing countries and not analyze the others? Any explanation is necessary.

Thirdly, section 5.1 should be merged into section 5. Usually, the journal does not have a subsection in the conclusion part. 

Finally, does the author have any further research directions in the future based on existing research? This is also what the authors need to explicitly list in the conclusion section.

I hope the above suggestions are usfult.

Happy New Year.

Author Response

Thanks for your constructive comments and suggestions, which have helped us to develop the paper further. We have taken all your comments and suggestions on board. Where we were unable to address the comments, we have provided justification for not incorporating the comments. We reproduce each of your original comments in italics followed in turn by our responses.

Dear author(s),

I think the logical chain of the paper itself is clear. However as a paper to be published in the JCR journal, it seems that the article lacks some depth, so I suggest that the author make the following changes.

  1. First of all, what is the significance of the article making such a statistical analysis? Is it to highlight policy recommendations for economic development in developing countries? We have seen a lot of detailed statistics, but we have not seen this part of the discussion, so I think the author needs to add the relevant part of the theoretical discussion.

The theoretical discussion section has been added into the paper. The results of this study have been explained with the help of the theory [see lines 214-276].

  1. I would like to suggest the following changes. Economic theories show that entrepreneurs are the driving force of the market economy, not the government. The authors should emphasize how the entrepreneurs play their role in the market economy and how this could affect CSR.

I recommend the authors to refer to the following books and articles and provide a structural section on entrepreneurship, economic development CSR. Of course, apart from the following references, the authors can also address other theoretical works that they know and that is related to the above topics.

Huerta de Soto, J. (2010). Socialism, Economic Calculation and Entrepreneurship. Edward Elgar.

Foss, N. J., & Klein, P. G. (2012). Organizing entrepreneurial judgment: A new approach to the firm. Cambridge University Press.

Harper, D. A. (2003). Foundations of entrepreneurship and economic development. Routledge.

Espinosa, V. I., Wang, W. H., & Zhu, H. (2020). Israel Kirzner on dynamic efficiency and economic development. Procesos de mercado: revista europea de economía política, 17(2), 283-310.

Since this research focuses on the determinants/motivations of CSR disclosure in developing countries. This point seems does not fall in the scope of this paper. Therefore, this point could not be addressed in this research.

  1. Another question is, why choose these developing countries and not analyze the others? Any explanation is necessary.

This paper was written for a special issue of sustainability, targeting CSR in developing countries. Therefore, this paper studied CSR disclosure studies conducted in developing countries only.

  1. Thirdly, section 5.1 should be merged into section 5. Usually, the journal does not have a subsection in the conclusion part. 

This point has been addressed by merging section 5.1 into section 5 [see lines 123-130].

  1. Finally, does the author have any further research directions in the future based on existing research? This is also what the authors need to explicitly list in the conclusion section.

This point has already been addressed and the future research directions have been mentioned in the conclusion section of this paper [see lines 296-312].

I hope the above suggestions are usfult.

Happy New Year.

Looking forward for positive reply.

Kind Regards,

Waris Ali, Jeffery Wilson, and Muhammad Husnain

Round 2

Reviewer 2 Report

The authors introduced some of my comments.

However, I urge you to clearly state the purpose of the article in the abstract by saying "The purpose of the article is ...". It is important for me.

I believe that to the 71 analyzed studies it is worth adding one more for Poland: https://www.researchgate.net/publication/340914864_Doubts_of_the_TSL_Enterprises_to_Social_Responsibility_-_An_Empirical_Study_Based_on_the_Results_of_Research

It shows the attitude of logistics companies to CSR. 

Author Response

Dear Reviewer,

Thanks for your constructive comments and suggestions, which have helped us to develop the paper further. We have taken all your comments and suggestions on board. We reproduce each of your original comments in italics followed in turn by our responses.

  1. However, I urge you to clearly state the purpose of the article in the abstract by saying "The purpose of the article is ...". It is important for me.

The point has been addressed. The purpose of the study has clearly been mentioned in the abstract [see line 9-10]

  1. I believe that to the 71 analyzed studies it is worth adding one more for Poland: https://www.researchgate.net/publication/340914864_Doubts_of_the_TSL_Enterprises_to_Social_Responsibility_-_An_Empirical_Study_Based_on_the_Results_of_Research It shows the attitude of logistics companies to CSR. 

This point has been addressed. The suggested study has been incorporated into the article [see lines 221-223].

Thanks for favorable comments and we appreciate your valuable comments.

Kind Regards,

Waris Ali, Jeffery Wilson, and Muhammad Husnain

Reviewer 3 Report

I accept the article for publication, the modified version can be published

Author Response

Dear Reviewer,

Thanks for your constructive comments and suggestions, which have helped us to develop the paper further. We have taken all your comments and suggestions on board. We reproduce each of your original comments in italics followed in turn by our responses.

  1. I accept the article for publication, the modified version can be published

We appreciate your valuable comments and accepting the paper for publication

Kind Regards,

Waris Ali, Jeffery Wilson, and Muhammad Husnain

Reviewer 4 Report

Dears authors,

Thank you for your modificaiton. However, there are still some issues that should be solved.

It is unnecessary to have a subsection in the conclusion part, section 6.1 should be integrated into section 6 as the conclusion part.

I've noticed your new modified lines [214-276] do talk about economic development and entrepreneurship. While the references like the recommended are necessary to de demonstrated as they do deal with the issues of developing countries and firms. It is not necessary to make a resume of the recommended references, but the references of the same category are very necessary to make the paper has a better theoretical and sound ground. Currently, I am concerned that it is still missed here.

Thank you.

Best wishes.

Author Response

Dear Reviewer,

Thanks for your constructive comments and suggestions, which have helped us to develop the paper further. We have taken all your comments and suggestions on board. We reproduce each of your original comments in italics followed in turn by our responses.

  1. It is unnecessary to have a subsection in the conclusion part, section 6.1 should be integrated into section 6 as the conclusion part.

This point has been addressed by integrating the section 6 with the conclusion part [see lines 272-307].

  1. I've noticed your new modified lines [214-276] do talk about economic development and entrepreneurship. While the references like the recommended are necessary to de demonstrated as they do deal with the issues of developing countries and firms. It is not necessary to make a resume of the recommended references, but the references of the same category are very necessary to make the paper has a better theoretical and sound ground. Currently, I am concerned that it is still missed here.

This point has been addressed. We have added a small paragraph on the role of entrepreneurship and cited the suggested studies [see lines 304-308]

Looking forward for positive reply.

Kind Regards,

Waris Ali, Jeffery Wilson, and Muhammad Husnain

Round 3

Reviewer 2 Report

The authors have revised their article as noted in the review. In general, the article has been improved in many places, which increased its substantive value. I recommend the article for publication. 

Author Response

Dear Reviewer,

Thank you for your kind words and acceptance of our paper for publication.

Waris Ali, Jeffery Wilson, and Muhammad Husnain

Reviewer 4 Report

I think this version is much better. Thank you for your effort. However, there are still some fields that the author(s) might improve.

First of all, the authors should address the significance of CSR directly in the abstract. Otherwise, it is difficult to let the readers know clearly the contribution that the paper can make.

Second, due to lines 309-310. The authors say that none of the 71 studies that the authors analyzed have discussed the role of entrepreneurship. I suggest that the authors might check it again. Indeed economists differ in their opinion. However, it is quite difficult to find any economist who is completely against entrepreneurship or does not address it. Entrepreneurship is also related to the firm's organizations. The authors might check the studies from this field.

Thirdly, I think the authors have improved a lot the part of the limitation of this paper in the conclusion part and propose further research suggestions due to various reviewers, which is good. In this regard, I would suggest the authors also show the research limitation issues directly in the methodology part to make the general structure of the paper more sound.

Best.

Author Response

Dear Reviewer,

Thank you for your constructive comments and suggestions, which have assisted us in further developing the paper. We have considered all of your comments and suggestions. Each of your original comments is reproduced in italics, followed by our responses.

  1. I think this version is much better. Thank you for your effort. However, there are still some fields that the author(s) might improve.

Thank you very much for your acknowledgement and appreciation.

  1. First of all, the authors should address the significance of CSR directly in the abstract. Otherwise, it is difficult to let the readers know clearly the contribution that the paper can make.

This point has been addressed by including a few lines in the abstract about the importance of CSR disclosure research [please see lines 9-13].

  1. Second, due to lines 309-310. The authors say that none of the 71 studies that the authors analyzed have discussed the role of entrepreneurship. I suggest that the authors might check it again. Indeed economists differ in their opinion. However, it is quite difficult to find any economist who is completely against entrepreneurship or does not address it. Entrepreneurship is also related to the firm's organizations. The authors might check the studies from this field.

The authors attempted to address this issue by emphasizing the role that corporate entrepreneurial orientation plays in shaping CSR activities [please see lines 305-3016]. The authors were unable to locate a single empirical study published in ABS-ranked journals that investigated the relationship between entrepreneurship and CSR disclosure in developing countries. As a result, the authors have proposed the same in terms of future research directions. I hope this addresses the reviewer's concern.

  1. Thirdly, I think the authors have improved a lot the part of the limitation of this paper in the conclusion part and propose further research suggestions due to various reviewers, which is good. In this regard, I would suggest the authors also show the research limitation issues directly in the methodology part to make the general structure of the paper more sound.

This point has already been incorporated in the methodology part of this paper [please see lines 83-84]

Looking forward for positive reply.

Kind Regards,

Waris Ali, Jeffery Wilson, and Muhammad Husnain
